# GDLNN: Marriage of Programming Language and Neural Networks for Accurate and Easy-to-Explain Graph Classification

## Abstract

We present GDLNN, a new graph machine learning architecture, for graph classification tasks. GDLNN combines a domain-specific programming language, called GDL, with neural networks. The main strength of GDLNN lies in its GDL layer, which generates expressive and interpretable graph representations. Since the graph representation is interpretable, existing model explanation techniques can be directly applied to explain GDLNN's predictions. Our evaluation shows that the GDL-based representation achieves high accuracy on most graph classification benchmark datasets, outperforming dominant graph learning methods such as GNNs. Applying an existing model explanation technique also yields high-quality explanations of GDLNN's predictions. Furthermore, the cost of GDLNN is low when the explanation cost is included.

## 1 Introduction

In graph classification, graph representation learning holds the key to success. The goal of this task is to learn useful feature representations (embeddings) for the entire graph that effectively capture its key structure and properties. These representations are utilized in various graph machine learning tasks, including graph classification. Beyond predictive performance, learning interpretable graph representations has become increasingly important, as it directly impacts model explainability, crucial in decision-critical domains such as drug discovery (Kakkad et al., 2023). Despite the dual requirement of both accuracy and explainability, generating effective and interpretable graph representations for entire graphs remains a key challenge.

***Current Trends and Limitations.*** Currently, the dominant approach to graph representation learning for graph-level tasks is the combination of graph neural networks (GNNs) and graph pooling operations (Wu et al., 2021). GNNs perform message passing to generate node and edge representations that capture local structures. After message passing, a pooling operation aggregates the node- and edge-level representations into a single graph-level representation (Grattarola et al., 2024).

During this two-step process, the resulting graph-level representation loses substantial structural information, which degrades both accuracy and explainability. Because the graph-level representation is a complex mixture of all node and edge features, important structural details are often obscured or lost during aggregation. This loss restricts the model's ability to capture key patterns crucial for accurate predictions, while also limiting its ability to explain graph classification results. For example, standard model explanation techniques that identify key features of the representation are inapplicable for explaining GNNs (Yuan et al., 2022). As a result, GNNs are explained using inefficient and indirect methods (Jeon et al., 2024).

***Our Approach.*** To address these problems, we present GDLNN (Graph Description Language + Neural Network), a novel architecture that generates interpretable graph representations that preserve key graph structures. GDLNN leverages a domain-specific programming language, GDL (Graph Description Language (Jeon et al., 2024)), to explicitly capture key graph patterns as representations. During training, GDLNN mines a set of high-quality GDL programs that capture discriminative patterns across the training graphs. Each GDL program describes a specific graph pattern, enabling GDLNN to represent graphs as collections of such patterns through its GDL layer.

The resulting representation is then fed into a Multi-Layer Perceptron (MLP) for classification, combining the interpretability of programming languages with the predictive power of neural networks.

***Results.*** Our evaluation shows that GDLNN achieves high accuracy on most of graph classification benchmarks, and its predictions can be explained using standard model explanation techniques. For example, GDLNN consistently outperforms popular GNNs across these datasets. Combination of GDLNN with a standard model explanation technique LIME (Ribeiro et al., 2016) also achieves better explainability than GNNs with state-of-the-art GNN explanation techniques SUBGRAPHX (Yuan et al., 2021). In addition, GDLNN is efficient when explanation costs are taken into account.

***Contributions.*** We summarize our contributions as follows:

- We present GDLNN, a novel graph machine learning architecture that combines a domain-specific programming language (GDL) with neural networks (NN).
- We experimentally demonstrate that GDLNN achieves high accuracy on graph classification benchmarks and can provide high-quality explanations through applying existing model explanation techniques. Also, GDLNN is fast when the explanation cost is included.

## 2 RELATED WORKS

***Graph Neural Networks.*** Graph Neural Networks (GNNs) are dominant methods in graph machine learning because of their high accuracy. In graph classification, GNNs (Hamilton et al., 2017; Defferrard et al., 2016) perform a message-passing procedure to generate node and edge representations that capture local structure (i.e., neighborhood information). For instance, GCN (Kipf & Welling, 2017) is a simplified convolutional network that uses a localized first-order approximation of spectral graph convolutions. GAT (Veličković et al., 2018) learns to weigh the importance of neighboring nodes via attention mechanisms, enabling more relevant neighbors to contribute more when updating a node's representation. GIN (Xu et al., 2019) aggregates neighbor features using a sum operation and applies an MLP, making it as powerful as the Weisfeiler-Lehman graph isomorphism test in distinguishing graph structures. After message passing, GNNs apply a pooling mechanism to aggregate node representations into a single representation of the entire graph (Grattarola et al., 2024). During this two-step process, the graph-level representation loses significant structural information. This not only degrades accuracy but also makes the representation difficult to interpret; the predictions of GNNs are challenging to explain (Kakkad et al., 2023).

***Explainable Graph Machine Learning.*** To achieve both accuracy and explainability, several graph machine learning methods have been proposed. A mainstream approach is to explain the predictions of GNNs (Kakkad et al., 2023). For instance, GraphChef (Müller et al., 2023) aims to interpret a GNN model by learning a decision tree that is faithful to it. SubgraphX (Yuan et al., 2021) generates subgraphs as explanations for GNN predictions. Inherently explainable graph learning methods have also been explored. For example, PL4XGL (Jeon et al., 2024) is an inherently explainable graph learning method that employs GDL programs for classification. PL4XGL classifies a graph using the highest-weighted GDL program and provides it as an explanation. However, existing explainable graph machine learning methods face limitations: inherently explainable approaches often suffer from low accuracy, while explaining GNN predictions is computationally costly and may not be faithful. GDLNN addresses these limitations by combining a domain-specific programming language with neural networks.

## 3 OVERVIEW

This section provides a high-level overview of how GDLNN works using a running example.

***Example graphs.*** Figures 1a, 1b, 1c, and 1d depict four example directed graphs. Each graph contains four nodes, each associated with a one-dimensional feature ranging from 1.0 to 4.0. In $G_1$, for example, the successor of the node with feature value $\langle 4.0 \rangle$ has a feature value of $\langle 2.0 \rangle$. Graphs $G_1$ and $G_3$ are labeled 1, while graphs $G_2$ and $G_4$ are labeled 2. Our goal is to generate graph representations such that graphs with the same label have similar representations.

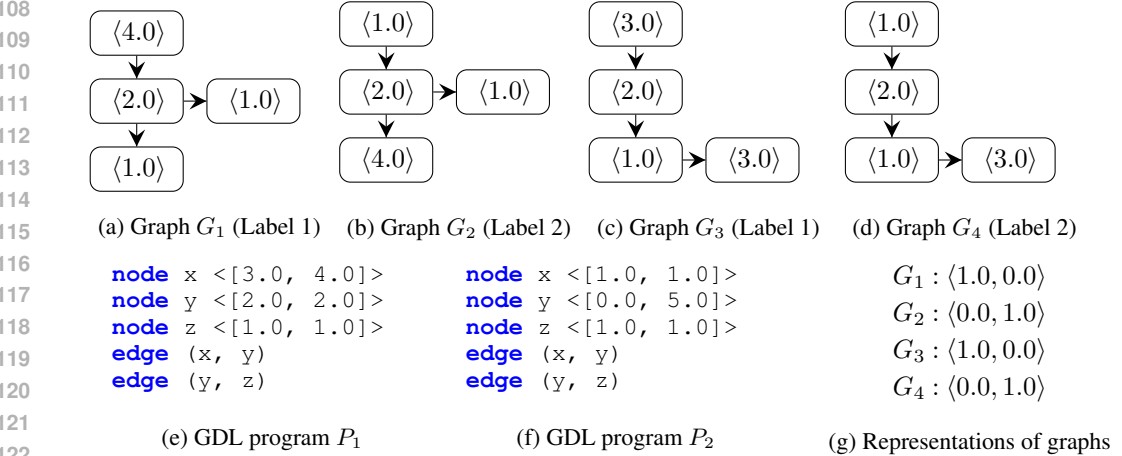

(a) Graph $G_1$ (Label 1)   (b) Graph $G_2$ (Label 2)   (c) Graph $G_3$ (Label 1)   (d) Graph $G_4$ (Label 2)

```
node x <[3.0, 4.0]>        node x <[1.0, 1.0]>
node y <[2.0, 2.0]>        node y <[0.0, 5.0]>
node z <[1.0, 1.0]>        node z <[1.0, 1.0]>
edge (x, y)                edge (x, y)
edge (y, z)                edge (y, z)
```

$G_1 : \langle 1.0, 0.0 \rangle$

$G_2 : \langle 0.0, 1.0 \rangle$

$G_3 : \langle 1.0, 0.0 \rangle$

$G_4 : \langle 0.0, 1.0 \rangle$

(e) GDL program $P_1$          (f) GDL program $P_2$          (g) Representations of graphs

Figure 1: An example showing how GDL programs are used for generating graph representations.

***How GDLNN works.*** Figure 1e and 1f show two example GDL programs that are used for generating two-dimensional graph representations. A GDL program consists of node and edge variables, each associated with a vector of feature value constraints. For example, the GDL program $P_1$ in Figure 1e contains three node variables ($x$, $y$, $z$) and two edge variables (($x$, $y$) and ($y$, $z$)). The node variables $x$, $y$, and $z$ are associated with vectors of feature value constraints $\langle [3.0, 4.0] \rangle$, $\langle [2.0, 2.0] \rangle$, and $\langle [1.0, 1.0] \rangle$, respectively. In English, the GDL program $P_1$ describes the following graph pattern:

> *"Graphs that contain a node with a feature value between 3.0 and 4.0, whose successor has a feature value of 2.0, and the successor also has a successor with a feature value of 1.0."*

For example, graphs $G_1$ and $G_3$ are described by the GDL program $P_1$, since a subgraph of each graph (e.g., ⟨4.0⟩ → ⟨2.0⟩ → ⟨1.0⟩ in $G_1$) matches the pattern defined by $P_1$. In contrast, graphs $G_2$ and $G_4$ are not described by $P_1$. The GDL program $P_2$ in Figure 1f captures a different graph pattern, described as follows:

> *"Graphs that contain a node with a feature value of 1.0, whose successor has a feature value between 0.0 and 5.0, and the successor also has a successor with a feature value 1.0."*

$G_2$ (Figure 1b) and $G_4$ (Figure 1d) match this pattern, as their subgraphs ( ⟨1.0⟩ → ⟨2.0⟩ → ⟨1.0⟩ ) satisfy the constraints defined by $P_2$. In contrast, graphs $G_1$ and $G_3$ do not match this pattern.

Figure 1g illustrates the two-dimensional graph representations of the four graphs, generated using the GDL programs $P_1$ and $P_2$. Each graph representation is a vector indicating whether the corresponding graph is described by each of the two GDL programs. For example, graphs $G_1$ and $G_3$ have representations $\langle 1.0, 0.0 \rangle$, since they are described by $P_1$ but not by $P_2$.

Once the graph representations are generated, GDLNN uses a multi-layer perceptron (MLP) to classify the graphs. That is, the performance of GDLNN depends on both the quality of the learned GDL programs and the effectiveness of the MLP. During training, GDLNN learns a set of high-quality GDL programs that would generate similar representations for graphs with the same label, and trains the MLP to accurately classify the graphs based on the generated representations.

***Explaining Classifications.*** As the graph representations used in GDLNN are interpretable, existing model explanation techniques can be applied directly. For example, LIME (Ribeiro et al., 2016) or SHAP (Lundberg & Lee, 2017) can be applied to identify the most important features of the graph representation, thereby explaining the classification results. For instance, suppose LIME determines that the first feature of the graph representation is important when GDLNN classifies graph $G_1$ as label 1. In this case, a user can identify that the GDL program $P_1$ corresponds to a key graph pattern driving the classification in GDLNN.

$$
\begin{array}{llllll}
\text{Programs} & P & ::= \overline{\delta} & \in & \mathbb{P} & = \mathbb{D}^* \\
\text{Descriptions} & \delta & ::= \delta_{\mathcal{V}} \mid \delta_{\mathcal{E}} & \in & \mathbb{D} & = \mathbb{D}_{\mathcal{V}} \uplus \mathbb{D}_{\mathcal{E}} \\
\text{Node Descriptions} & \delta_{\mathcal{V}} & ::= \textbf{node } x \texttt{<}\overline{\phi}\texttt{>}^? & \in & \mathbb{D}_{\mathcal{V}} & = \mathbb{X} \times \Phi^d \\
\text{Edge Descriptions} & \delta_{\mathcal{E}} & ::= \textbf{edge } (x,x) \texttt{<}\overline{\phi}\texttt{>}^? & \in & \mathbb{D}_{\mathcal{E}} & = \mathbb{X} \times \mathbb{X} \times \Phi^c \\
\text{Intervals} & \phi & ::= [r,r] & \in & \Phi & = (\mathbb{R} \cup \{-\infty\}) \times (\mathbb{R} \cup \{\infty\}) \\
\text{Real Numbers} & r & ::= \texttt{0.2} \mid \texttt{0.7} \mid \texttt{6} \mid \texttt{-8} \dots & \in & \mathbb{R} & \\
\text{Variables} & x & ::= \texttt{x} \mid \texttt{y} \mid \texttt{z} \mid \dots & \in & \mathbb{X} &
\end{array}
$$

Figure 2: The syntax of GDL.

$$
\begin{array}{ll}
[\![\texttt{<}\phi_1, \dots, \phi_k\texttt{>}]\!] & = \{ \quad \mathbf{x} \quad \mid \mathbf{x} = \langle \mathbf{x}_1, \dots, \mathbf{x}_k \rangle \wedge \forall i.\ \phi_i = [a,b] \Rightarrow a \leq \mathbf{x}_i \leq b \} \\
[\![\textbf{node } x \texttt{<}\overline{\phi}\texttt{>}]\!] & = \{ (G, \eta) \mid v = \eta(x) \wedge X_i^{\mathcal{V}} \in [\![\texttt{<}\overline{\phi}\texttt{>}]\!] \} \\
[\![\textbf{edge } (x,y) \texttt{<}\overline{\phi}\texttt{>}]\!] & = \{ (G, \eta) \mid e \in \mathcal{E} \wedge e = (\eta(x), \eta(y)) \wedge X_j^{\mathcal{E}} \in [\![\texttt{<}\overline{\phi}\texttt{>}]\!] \} \\
[\![\delta_1 \delta_2 \dots \delta_k]\!] & = \{ (G, \eta) \mid \forall i.\ (G, \eta) \in [\![\delta_i]\!] \}
\end{array}
$$

Figure 3: The semantics of GDL where $G = (\mathcal{V}, \mathcal{E}, X^{\mathcal{V}}, X^{\mathcal{E}})$ is a graph.

# 4 GDLNN

In this section, we formally define GDLNN. We first introduce GDL, a domain-specific programming language for describing graph patterns, and then present the architecture of GDLNN.

***Notations.*** We consider a directed graph $G = (\mathcal{V}, \mathcal{E}, X^{\mathcal{V}}, X^{\mathcal{E}})$, where $\mathcal{V} = \{v_1, v_2, \dots, v_n\}$ is the set of nodes, and $\mathcal{E} = \{e_1, e_2, \dots, e_m\} \subseteq \mathcal{V} \times \mathcal{V}$ is the set of edges. $X^{\mathcal{V}} \subseteq \mathbb{R}^{n \times d}$ and $X^{\mathcal{E}} \subseteq \mathbb{R}^{m \times c}$ are matrices of node and edge feature vectors, respectively. $X_i^{\mathcal{V}}$ and $X_j^{\mathcal{E}}$ denote the feature vectors of node $v_i$ and edge $e_j$, respectively.

## 4.1 GDL: GRAPH-PATTERN DESCRIPTION LANGUAGE

***Syntax of GDL.*** Figure 2 illustrates the syntax of GDL. $\overline{A}$ denotes a sequence of elements in $A$ (e.g., $\overline{A} = \langle A_1, A_2, \dots, A_k \rangle$). A GDL program consists of a sequence of descriptions $\overline{\delta}$, where each description is either a node description ($\delta_{\mathcal{V}}$) or an edge description ($\delta_{\mathcal{E}}$). A node (resp., edge) description combines a variable (e.g., **node** x or **edge** (x, y)) with a vector of intervals (e.g., <[3.0, 4.0], [1.0, 2.0], . . .>) that specifies the node's (resp., edge's) feature values.

***Semantics of GDL.*** We now define the semantics of GDL programs (i.e., the patterns they describe). Figure 3 presents the formal semantics of GDL. Given a vector of intervals $\texttt{<}\phi_1, \dots, \phi_k\texttt{>}$, its semantics $[\![\texttt{<}\phi_1, \dots, \phi_k\texttt{>}]\!]$ defines a set of feature vectors in which the $i$-th feature value lies within the interval $\phi_i$. For example, $[\![\texttt{<}[3.0, 4.0], [1.0, 2.0]\texttt{>}]\!]$ defines the set of feature vectors $\{\langle 3.0, 1.0 \rangle, \dots, \langle 4.0, 2.0 \rangle\}$. A valuation $\eta \in \mathbb{X} \to \mathcal{V}$ maps each node variable in the GDL program to a distinct node in a given graph $G = (\mathcal{V}, \mathcal{E})$. That is, $(G, \eta) \in [\![\overline{\delta}]\!]$ indicates that a subgraph of $G$ is captured by the GDL program $P = \overline{\delta}$ via the valuation $\eta$. For example, the GDL program $P_1$ in Figure 1e captures a subgraph of $G_1$ through the following valuation $\eta$, which defines the subgraph $G_1'$:

$$
\eta = \{\texttt{x} \to \boxed{\langle 4.0 \rangle}, \texttt{y} \to \boxed{\langle 2.0 \rangle}, \texttt{z} \to \boxed{\langle 1.0 \rangle} \}, \quad G_1' = \boxed{\langle 4.0 \rangle} \to \boxed{\langle 2.0 \rangle} \to \boxed{\langle 1.0 \rangle}
$$

## 4.2 ARCHITECTURE OF GDLNN

Figure 4 shows the architecture of GDLNN, which consists of GDL layer and NN layers.

***GDL Layer.*** Each perceptron in the GDL layer is associated with a GDL program $P_i$ mined during the training phase. Details of the training phase are presented in Section 5. The output of the $i$-th perceptron is determined by an activation function $\sigma(P_i, G)$, defined as:

$$
\sigma(P_i, G) = \begin{cases} 1.0 & \text{if } G \models P_i \\ 0.0 & \text{otherwise} \end{cases}
$$

where $G \models P_i$ indicates that the graph $G$ satisfies the pattern defined by $P_i$:

$$
G \models P \iff \exists \eta.\ (G, \eta) \in [\![P]\!].
$$

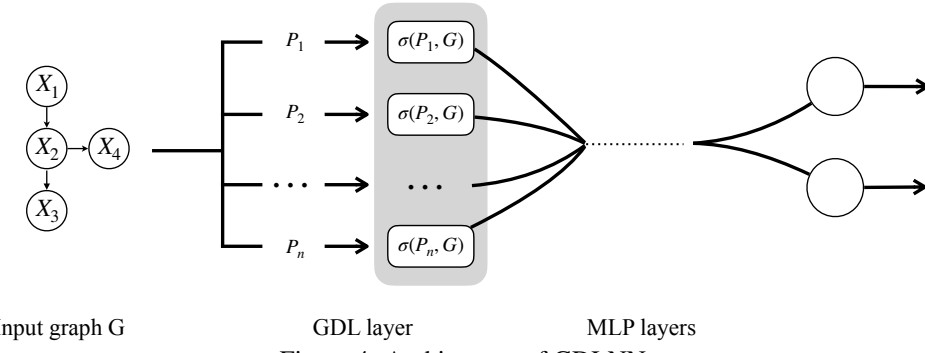

Figure 4: Architecture of GDLNN

---

**Algorithm 1** Overall GDL program mining procedure

---

**Require:** A training data $\mathcal{D} = \{(G_i, y_i)\}_{i=1}^m$
**Ensure:** GDL programs $\mathcal{P}$
1: **procedure** LEARN($\mathcal{D}$)
2:    $\mathcal{P} \leftarrow \{\}$
3:    **for each** $(G, y) \in \mathcal{D}$ **do**
4:        $P \leftarrow \text{MINE}_{Score}(\mathcal{D}, G, y)$
5:        $\mathcal{P} \leftarrow \mathcal{P} \cup \{P\}$
6:    **return** $TopK(\mathcal{P})$

---

Note that the activation function $\sigma$ is a design choice. Due to space limitations, alternative choices are discussed in Appendix A.

***MLP Layers.*** The MLP layers consist of standard multi-layer perceptrons. The MLP takes the generated graph representation as input and produces a classification output for the graph.

## 5    LEARNING GDLNN

Now we present the learning procedure of GDLNN. The learning procedure of GDLNN consists of two sequential steps: 1) mining GDL programs and 2) training the MLP. Let $\mathcal{D} = \{(G_i, y_i)\}_{i=1}^m$ be a training dataset (consists of $m$ graphs) where each $G_i$ represents a graph and $y_i \in L$ is its corresponding label ($L$ is the set of labels). For convenience, we define $\mathcal{D}_y = \{(G_i, y_i) \mid (G_i, y_i) \in \mathcal{D}, y_i = y\}$ as the training graphs in $\mathcal{D}$ with label $y$.

### 5.1    MINING GDL PROGRAMS

The learning procedure first mines GDL programs that will be used in the GDL layer.

***Objective.*** The objective of this procedure is to mine GDL programs that generate high-quality graph representations. To this end, our algorithm mines GDL programs that describe graphs having the same labels while excluding graphs with different labels. We expect that such mined GDL programs will produce high-quality graph representations. The detailed objective is presented in Appendix B.

***GDL Program Mining.*** Algorithm 1 presents the overall GDL program mining procedure. The procedure takes a training dataset $\mathcal{D} = \{(G_i, y_i)\}_{i=1}^m$ as input and outputs a set of GDL programs $\mathcal{P}$. Lines 3–5 describe the process of mining and collecting a GDL program from each training graph. At line 4, a GDL program $P$ is mined using the procedure $\text{MINE}_{Score}(\mathcal{D}, G, y)$. We employ an existing GDL program mining algorithm (Jeon et al., 2024), which aims to mine a GDL program $P$ that maximizes the quality score function *Score* defined as:

$$Score(P, \mathcal{D}, y) = \frac{|\{G \mid (G, y) \in \mathcal{D}_y, (G, \eta) \in [\![P]\!] \}|}{|\{G \mid (G, y) \in \mathcal{D}, (G, \eta) \in [\![P]\!] \}| + \epsilon}$$

---

**Algorithm 2** Explaining GDLNN

---

**Require:** graph $G = (\mathcal{V}, \mathcal{E}, X^{\mathcal{V}}, X^{\mathcal{E}})$, GDLNN model $\mathcal{M}$, explanation method $T$
**Ensure:** subgraph explanation $G'$
 1: **procedure** EXPLAIN($G, \mathcal{M}, T$)
 2:     $\mathcal{P} \leftarrow ImportantFeatures(G, \mathcal{M}, T)$
 3:     $G' \leftarrow G$
 4:     **repeat**
 5:         $G \leftarrow G'$
 6:         $G' \leftarrow Refine(G, \mathcal{P})$
 7:     **until** $G = G'$
 8:     **return** $G'$

---

where $\epsilon$ is a hyperparameter. Intuitively, $Score(P, \mathcal{D}, y)$ measures the precision of the mined GDL program $P$ in describing graphs with label $y$ in the training dataset $\mathcal{D}$. $\epsilon$ enables the score to consider robustness of the mined GDL program. Using a bigger $\epsilon$ makes robust programs (i.e., describe many graphs that belong to label $y$) have a higher score than others. In our evaluation we set $\epsilon$ as 0.1, 1, or $|\mathcal{D}| * 0.01$ that performs best on the validation set. Due to the space limit, the detailed mining algorithm is presented in Appendix B. Each mined program is collected into $\mathcal{P}$ at line 5. At line 6, the algorithm selects the top-$k$ GDL programs based on their scores:

$$TopK(\mathcal{P}) = \{P \in \mathcal{P} \mid |\{P' \in \mathcal{P} \mid Score(P', \mathcal{D}, y) \geq Score(P, \mathcal{D}, y)\}| \leq k\}$$

where $k$ is a hyperparameter that determines the number of units in the GDL layer. We select $k$ from $\{0.01 \cdot |\mathcal{D}|, 0.2 \cdot |\mathcal{D}|, 0.4 \cdot |\mathcal{D}|, 0.6 \cdot |\mathcal{D}|, 0.8 \cdot |\mathcal{D}|, |\mathcal{D}|\}$ based on the validation set.

***MLP Training.*** After mining GDL programs, the MLP layers are trained to classify graphs based on the representations generated by the GDL layer. The MLP takes these graph representations as input and outputs the predicted label $\hat{y}$.

## 5.2 EXPLAINING GDLNN

The predictions of GDLNN can be explained by directly applying existing model explanation techniques. In GDLNN, each feature corresponds to a graph pattern (i.e., a GDL program); thus, identifying key features directly reveals the graph patterns responsible for the prediction.

The identified key features can also be transformed into subgraph explanations. In the graph machine learning domain, explanations are usually subgraphs (Kakkad et al., 2023) describing key parts that are responsible for the predictions. Algorithm 2 illustrates how to generate subgraph explanations in GDLNN using existing model explanation techniques that identify important features. The algorithm takes as input a target graph $G$ (to be explained), a GDLNN model $\mathcal{M}$, and an explanation method $T$. At line 2, the algorithm collects the important features (i.e., GDL programs) using the explanation technique $T$. It then iteratively refines the graph into a subgraph that preserves the same feature values for the important features $\mathcal{P}$. At line 6, $Refine(G, \mathcal{P})$ attempts to find a subgraph $G'$ by removing nodes (and their corresponding edges) from $G$, such that:

$$\forall P \in \mathcal{P}.G \models P \Rightarrow G' \models P.$$

If no such subgraph exists, $Refine(G, \mathcal{P})$ returns the original graph $G$. The algorithm then outputs this subgraph as the explanation.

## 6 EVALUATION

In this section, we evaluate the performance of GDLNN on various graph classification benchmarks. Our evaluation aims to answer the following research questions:

- **RQ1 (Accuracy):** How does GDLNN compare to existing popular graph machine learning methods in terms of classification accuracy?

- **RQ2 (Explainability):** How does the explainability of GDLNN compare to other methods?

- **RQ3 (Cost):** How does the computational cost of GDLNN compare to baseline methods?

Table 1: Accuracy comparison on nine graph classification datasets.

|  | MUTAG | Mutagenicity | BBBP | BACE | ENZYMES | PROTEINS | PTC | NCI1 | BA-2Motifs |
|---|---|---|---|---|---|---|---|---|---|
| GIN | 94.0±5.1 | 81.0±1.0 | 82.9±3.3 | 81.1±3.8 | 64.3±3.4 | 71.3±3.9 | 61.2±7.4 | 81.3±2.4 | 99.8±0.5 |
| GCN | 72.0±13.6 | 79.6±1.7 | 83.0±1.7 | 74.3±17.8 | 65.9±3.0 | **71.7±1.6** | 54.1±4.1 | 77.4±0.8 | 99.8±0.5 |
| GAT | 90.0±0.0 | 76.0±1.2 | 84.4±2.8 | 65.5±15.4 | 60.0±5.2 | 72.8±4.5 | 62.3±12.5 | 65.8±3.3 | 81.2±4.4 |
| MLP | 93.0±10.3 | 80.1±1.6 | 83.4±3.4 | 81.3±3.1 | 54.3±6.9 | **71.7±2.5** | 49.4±9.1 | 77.8±1.2 | 99.8±0.5 |
| PL4XGL | **100±0.0** | 76.9±0.0 | 86.3±0.0 | 82.8±0.0 | 50±0.0 | 62.1±0.0 | 67.6±0.0 | 72.7±0.0 | **100.0±0.0** |
| GDLNN | **100±0.0** | **81.1±1.1** | **88.2±1.1** | **84.4±1.0** | **68.3±2.6** | 68.1±1.2 | **68.8±6.1** | **83.3±1.2** | **100.0±0.0** |

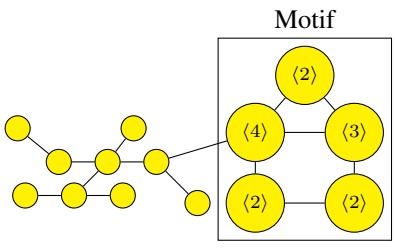

(a) An example graph of label 1

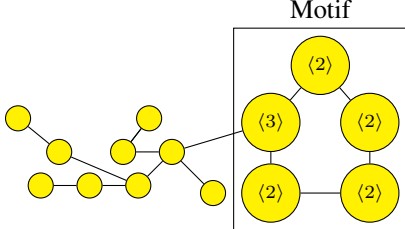

(b) An example graph for label 2

Figure 5: Example graphs in BA-2Motifs dataset.

**Datasets.** We evaluate GDLNN on nine widely used graph classification datasets. We use eight molecular datasets: BBBP, BACE, Mutagenicity, ENZYMES, PROTEINS, MUTAG, PTC (MR), and NCI1. Additionally, we include a synthetic dataset, BA-2Motifs, which is commonly used for evaluating explainability (Yuan et al., 2021). Dataset statistics are provided in Appendix C.

**Baselines.** For comparison, we include five popular baseline methods representing different approaches to graph classification. First, we consider three widely-used graph neural networks: GCN (Kipf & Welling, 2017), GIN (Xu et al., 2019), and GAT (Veličković et al., 2018). We also include MLP, a simple multi-layer perceptron that generates graph representations via graph pooling, and PL4XGL (Jeon et al., 2024), a symbolic (non-neural) graph learning method that employs GDL programs for classification. Hyperparameters are selected based on the validation set. The hyperparameter search space is provided in Appendix D. All experiments are conducted on an AMD Ryzen Threadripper 3990X (64 cores) with an NVIDIA RTX A6000 GPU. Datasets are randomly split into 80/10/10 for training, validation, and test sets.

### 6.1 ACCURACY COMPARISON

We first compare the accuracy of GDLNN with the baseline graph machine learning methods. Following prior work (Park et al., 2022), we report the 95% confidence intervals over five runs. Table 1 presents the comparison results, with the best-performing models for each dataset highlighted in bold. As shown, GDLNN achieves the highest accuracy on all datasets except PROTEINS. Compared to the baseline GNNs, GDLNN consistently shows better accuracy, demonstrating that GDL-based graph representations are a competitive alternative to standard message-passing and pooling-based embeddings for high accuracy. Compared to PL4XGL, a symbolic graph learning method using GDL programs, GDLNN achieves higher accuracy across all datasets. This improvement is mainly due to the neural network layers in GDLNN, which can capture more complex patterns in the graph data. Compared to MLP, a simple multi-layer perceptron with graph pooling, GDLNN also performs consistently better on all datasets except PROTEINS. This shows that pooled node features are useful for the PROTEINS dataset, and relying solely on GDL program-based features may slightly reduce accuracy. Developing an ensemble of GDL-based and pooled node features is a promising direction for future work to further improve accuracy.

### 6.2 EXPLAINABILITY COMPARISON

Next, we compare the explainability of GDLNN with baseline methods. In our evaluation, we adopt LIME (Ribeiro et al., 2016) to explain GDLNN. Once LIME identifies the important features, we transform these features into subgraph explanations using Algorithm 2.

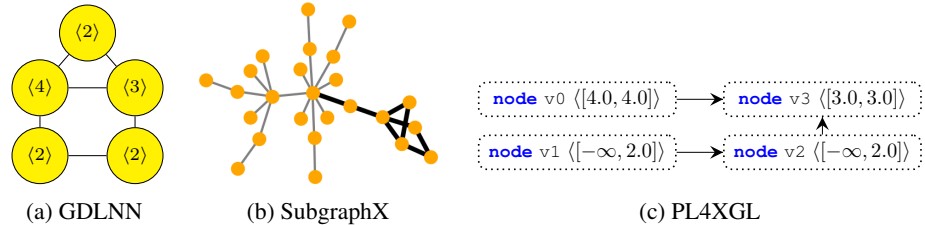

(a) GDLNN      (b) SubgraphX      (c) PL4XGL

Figure 6: Provided explanations for graphs classified into label 1

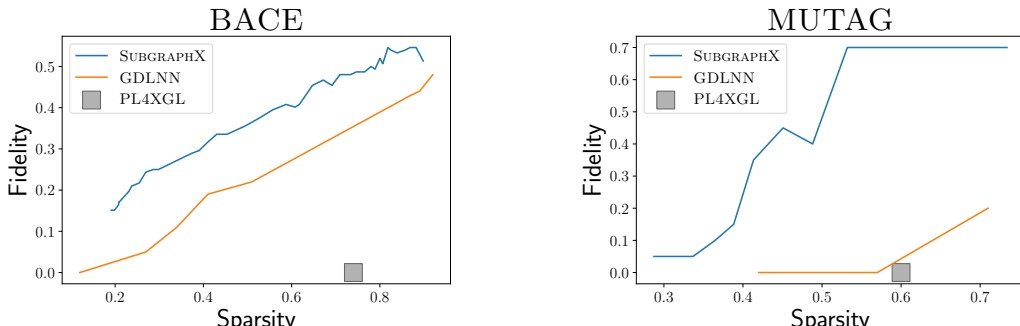

Figure 7: Quantitative comparison of explanations on the BACE and Mutagenicity datasets. A lower *Fidelity* value indicates that the explanations are more faithful to the model's predictions (i.e., lower is better). A higher *Sparsity* value indicates that the explanations are simpler (i.e., higher is better).

***Baselines.*** As baselines, we use SUBGRAPHX (Yuan et al., 2021), a state-of-the-art GNN explanation technique. Intuitively, SUBGRAPHX identifies important subgraphs that contribute to the classification decisions of GNNs. Following Yuan et al. (2021), we apply SUBGRAPHX to explain GIN's predictions. We also compare GDLNN with PL4XGL, an inherently explainable graph learning method. PL4XGL classifies a graph using one of the highest-scoring GDL programs and directly provides it as an explanation.

***Qualitative Comparison.*** We first qualitatively compare explanations using the synthetic dataset BA-2Motif, which is designed to evaluate the performance of model explanation techniques. In the dataset, each node is represented by its degree as a feature. For example, if a node has degree 3, its feature is $\langle 3.0 \rangle$. BA-2Motifs has two labels: label 1 and label 2. Each label is determined by the presence of a specific motif (i.e., graph pattern). The evaluation checks whether the explanation technique can correctly identify the motif. Figures 5a and 5b show example graphs from the BA-2Motifs dataset. Label 1 corresponds to a house-shaped motif consisting of five nodes (a roof node, two middle nodes, and two bottom nodes) connected by edges. One of the middle nodes is further connected to a randomly generated graph following the Barabási-Albert model. In the motif of label 1, the two middle nodes are connected. In contrast, the motif of label 2 also consists of five nodes, but the two middle nodes are not connected.

Figure 6 shows the explanations of GDLNN, SUBGRAPHX, and PL4XGL for graphs classified into label 1, and all three techniques correctly identify the motif-related property of label 1. For example, Figure 6a shows a subgraph explanation generated by GDLNN. The identified subgraph exactly matches the motif of label 1. Similarly, the subgraph explanation of SUBGRAPHX in Figure 6b (highlighted with bold edges) also contains the motif of label 1. PL4XGL provides a GDL program that captures a property describing the motif of label 1. Figure 6c shows a graphical representation of this GDL program, which consists of four node variables and three edge variables. The GDL program also correctly distinguishes graphs in label 1 from those in label 2. Explanations for label 2 similarly identified the motif of label 2. Explanations for label 2 are presented in Appendix E.

***Quantitative Comparison.*** Now, we quantitatively compare the explanations of GDLNN and SUBGRAPHX. For comparison, we use *Sparsity* and *Fidelity* as evaluation metrics (Kakkad et al., 2023). Intuitively, *Fidelity* measures the faithfulness of explanations; a lower score indicates higher faithful-

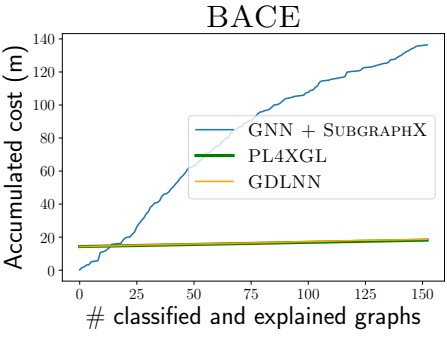 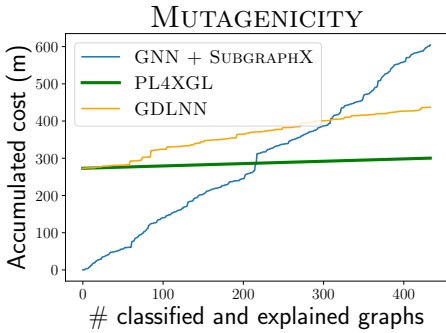

Figure 8: Accumulated (training + inference + explanation) cost comparison on the two datasets.

ness (i.e., lower is better). *Sparsity* measures the simplicity of explanations; a higher score indicates greater simplicity (i.e., higher is better). The formal definitions are provided in Appendix E.

Figure 7 compares the explainability of GDLNN(+LIME), GNN+SUBGRAPHX, and PL4XGL on the two metrics across two datasets. In SUBGRAPHX, the explanation size is user-defined; the blue lines indicate the performance of SUBGRAPHX. In GDLNN+LIME, the explanation size is determined by LIME (orange line). In our evaluation, LIME selected varying numbers of features, ranging from 5 to 1000. For PL4XGL, the explanation size is fixed by the model, and the gray squares indicate its performance. As the plots show, GDLNN achieves a better trade-off between sparsity and fidelity than SUBGRAPHX on both datasets. In terms of fidelity, GDLNN performs worse than PL4XGL because PL4XGL is inherently explainable (fidelity score is guaranteed to be 0 (Jeon et al., 2024)). By sacrificing some explainability, however, GDLNN achieves better accuracy than PL4XGL as shown in Table 1. We achieve consistent results for the other datasets. Due to space limits, results for the remaining seven datasets are provided in Appendix E.

### 6.3 COST OF GDLNN

Now, we compare the cost of GDLNN against the baselines. For fairness, we report the total accumulated cost, which includes training, inference, and explanation costs. Figure 8 presents the accumulated cost (in minutes) of GDLNN and the baselines on two datasets: BACE and Mutagenicity. In the figure, the y-axis shows the accumulated cost (training + inference + explanation), while the x-axis shows the number of classified and explained graphs. The blue lines correspond to GIN with SUBGRAPHX, and the green and orange lines correspond to PL4XGL and GDLNN(+LIME), respectively.

Compared to GIN+SUBGRAPHX, GDLNN is substantially faster. Initially (i.e., when only the training cost is included), GDLNN is more expensive than GNNs due to its costly GDL program mining process. However, the total accumulated cost of GDLNN becomes much lower than that of GNN+SUBGRAPHX, thanks to its significantly faster explanation process. Compared to PL4XGL, the cost difference is small since they share the same GDL learning process, and both have small classification and explanation costs. We also achieved consistent results for the other datasets. Due to space limits, results for the remaining seven datasets are provided in Appendix E.

## 7 CONCLUSION

In this paper, we present a new graph machine learning architecture called GDLNN, which is a combination of a domain-specific programming language and neural networks. Thanks to the effective and interpretable GDL program-based graph representations, GDLNN achieves outstanding accuracy on various graph classification benchmarks, and the predictions of GDLNN are easy to explain as the representations are inherently interpretable. Also, GDLNN is fast when the explanation cost is included. We believe this work is a starting point for a new graph machine learning architecture that employs effective and interpretable graph representations through domain-specific programming languages.

## 8 ETHICS STATEMENT

This work presents methods for graph classification and relies solely on publicly available benchmark datasets. To the best of our knowledge, these datasets do not contain personally identifiable information, and we have complied with their licenses and terms of use. No human-subject experiments, user studies, or data collection involving vulnerable populations were conducted in this work. Potential risks include harmful outcomes if learned GDL programs capture spurious correlations. Regarding privacy, since graph classification can be applied to sensitive domains such as social-network analysis, we discourage applications that enable surveillance, profiling, or other infringements of privacy and civil liberties.

## 9 REPRODUCIBILITY STATEMENT

We provide the source code of GDLNN and the datasets used in this work as a zip file in the supplementary material. The artifact provides the required environment and instructions to run the code. The manual is available as README.md in the supplementary material.

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

---

**Algorithm 3** GDL program mining algorithm

---

**Require:** A training data $\mathcal{D}$, a graph $G$, a label $y$, and a quality score function *Score*.
**Ensure:** GDL program $P$
1: **procedure** $\text{MINE}_{Score}(\mathcal{D}, G, y)$
2:      $P \leftarrow Initialize(G)$
3:      **while** *updated* **do**
4:          $\mathcal{P} \leftarrow Enumerate(P)$
5:          $P' \leftarrow Choose(\mathcal{P}, \mathcal{D}, y, Score)$
6:          **if** $Score(P, \mathcal{D}, y) > Score(P', \mathcal{D}, y) \vee P' = \bot$ **then**
7:             **return** $P$
8:          $P \leftarrow P'$

---

Petar Veličković, Guillem Cucurull, Arantxa Casanova, Adriana Romero, Pietro Liò, and Yoshua Bengio. Graph attention networks. In *International Conference on Learning Representations*, 2018.

Zonghan Wu, Shirui Pan, Fengwen Chen, Guodong Long, Chengqi Zhang, and Philip S. Yu. A comprehensive survey on graph neural networks. *IEEE Transactions on Neural Networks and Learning Systems*, 32(1):4–24, 2021. doi: 10.1109/TNNLS.2020.2978386.

Keyulu Xu, Weihua Hu, Jure Leskovec, and Stefanie Jegelka. How powerful are graph neural networks? In *7th International Conference on Learning Representations, ICLR 2019, New Orleans, LA, USA, May 6-9, 2019*. OpenReview.net, 2019. URL https://openreview.net/forum?id=ryGs6iA5Km.

Hao Yuan, Haiyang Yu, Jie Wang, Kang Li, and Shuiwang Ji. On explainability of graph neural networks via subgraph explorations. In *Proceedings of the 38th International Conference on Machine Learning (ICML)*, pp. 12241–12252, 2021.

Hao Yuan, Haiyang Yu, Shurui Gui, and Shuiwang Ji. Explainability in graph neural networks: A taxonomic survey. *IEEE Transactions on Pattern Analysis and Machine Intelligence*, pp. 1–19, 2022. doi: 10.1109/TPAMI.2022.3204236.

## A    ALTERNATIVE ACTIVATION FUNCTION

The activation function $\sigma$ in GDLNN is a design choice. Various alternative functions can be defined to generate values (i.e., embeddings) from a given graph $G$ and a GDL program $P_i$. For example, inspired by graphlets, we can define a more expressive activation function $\sigma'$ that counts the number of subgraphs in $G$ matching the pattern specified by $P_i$:

$$\sigma'(P_i, G) = |\{(G, \eta) \mid (G, \eta) \in [\![P_i]\!]\}|.$$

Unlike $\sigma$, which only indicates whether at least one subgraph of $G$ matches the pattern $P_i$, the function $\sigma'$ measures the number of distinct matching subgraphs. This provides richer information about $G$, though at the cost of higher inference time. In this work, we adopt the simple $\sigma$ function for efficiency, while exploring more sophisticated activation functions remains an interesting direction for future research.

## B    GDL PROGRAM MINING ALGORITHM

### B.1    OBJECTIVE

The goal of GDL program mining is to learn high-quality GDL programs that will be used in the GDL layer. The detailed objective of GDL program mining is to enable the GDL layer generate similar representations for graphs with the same labels while different representations for graphs with different labels. Formally, the goal is to mine a set of GDL programs $\mathcal{P} = \{P_1, \ldots, P_n\}$ that maximizes

$$\sum_{i,j} \text{sim}(\phi(G_i, \mathcal{P}), \phi(G_j, \mathcal{P})) \cdot (2\mathbb{I}[y_i = y_j] - 1)$$

where $\text{sim}(\phi(G_i, \mathcal{P}), \phi(G_j, \mathcal{P}))$ denotes the Hamming similarity between the representations of graphs $G_i$ and $G_j$ generated by the GDL programs $\mathcal{P}$. Here, $\mathbb{I}[y_i = y_j]$ is the indicator function, which returns 1 if $y_i = y_j$ and 0 otherwise. The representation $\phi(G, \mathcal{P})$ of a graph $G$ generated by the GDL programs $\mathcal{P}$ is defined as

$$\phi(G, \mathcal{P}) = \langle \sigma(P_1, G), \sigma(P_2, G), \ldots, \sigma(P_n, G) \rangle$$

A GDL layer consisting of programs $\langle P_1, \ldots, P_n \rangle$ that satisfies this objective is expected to yield highly discriminative representations, thereby achieving high accuracy on graph classification tasks.

## B.2 MINING A GDL PROGRAM

We reuse an existing GDL program mining algorithm (Jeon et al., 2024) to mine a GDL program $P$ that maximizes the quality score function *Score* defined in Section 5. Algorithm 3 presents the detailed procedure of the MINE function used in Algorithm 1. It takes a training dataset $\mathcal{D}$, a target training graph $G$, the label $y$ of the target graph $G$, and a quality score function *Score* as input and outputs a mined GDL program $P$. At line 2, the algorithm initializes a GDL program $P$ using the *Initialize* function with the given graph $G$. Suppose $\mathcal{V} = \{v_1, v_2, \ldots, v_n\}$ and $\mathcal{E} = \{e_1, e_2, \ldots, e_m\}$ are the node and edge sets of the graph $G$. Then, *Initialize*$(G)$ returns the following GDL program $P$ using a mapping function $mapNode : \mathcal{V} \to \mathbb{X}$ and $mapEdge : \mathcal{E} \to \mathbb{X} \times \mathbb{X}$:

> **node** $mapNode(v_1)$ <$[X^{\mathcal{V}}_{1,1}, X^{\mathcal{V}}_{1,1}], [X^{\mathcal{V}}_{1,2}, X^{\mathcal{V}}_{1,2}], \ldots, [X^{\mathcal{V}}_{1,d}, X^{\mathcal{V}}_{1,d}]$>
> **node** $mapNode(v_2)$ <$[X^{\mathcal{V}}_{2,1}, X^{\mathcal{V}}_{2,1}], [X^{\mathcal{V}}_{2,2}, X^{\mathcal{V}}_{2,2}], \ldots, [X^{\mathcal{V}}_{2,d}, X^{\mathcal{V}}_{2,d}]$>
> $\ldots$
> **node** $mapNode(v_n)$ <$[X^{\mathcal{V}}_{n,1}, X^{\mathcal{V}}_{n,1}], [X^{\mathcal{V}}_{n,2}, X^{\mathcal{V}}_{n,2}], \ldots, [X^{\mathcal{V}}_{n,d}, X^{\mathcal{V}}_{n,d}]$>
> **edge** $mapEdge(e_1)$ <$[X^{\mathcal{E}}_{1,1}, X^{\mathcal{E}}_{1,1}], [X^{\mathcal{E}}_{1,2}, X^{\mathcal{E}}_{1,2}], \ldots, [X^{\mathcal{E}}_{1,c}, X^{\mathcal{E}}_{1,c}]$>
> **edge** $mapEdge(e_2)$ <$[X^{\mathcal{E}}_{2,1}, X^{\mathcal{E}}_{2,1}], [X^{\mathcal{E}}_{2,2}, X^{\mathcal{E}}_{2,2}], \ldots, [X^{\mathcal{E}}_{2,c}, X^{\mathcal{E}}_{2,c}]$>
> $\ldots$
> **edge** $mapEdge(e_m)$ <$[X^{\mathcal{E}}_{m,1}, X^{\mathcal{E}}_{m,1}], [X^{\mathcal{E}}_{m,2}, X^{\mathcal{E}}_{m,2}], \ldots, [X^{\mathcal{E}}_{m,c}, X^{\mathcal{E}}_{m,c}]$>

where $mapNode$ maps each node to a distinct variable and $mapEdge$ maps each edge $(v_i, v_j)$ to a distinct variable pair $(mapNode(v_i), mapNode(v_j))$. $X^{\mathcal{V}}_{i,j}$ (resp., $X^{\mathcal{E}}_{i,j}$) denotes the $j^{\text{th}}$ value of the feature vector of node $X^{\mathcal{V}}_i$ (resp., edge $X^{\mathcal{E}}_i$). Intuitively, *Initialize*$(G)$ transforms the given graph $G$ into the most specific GDL program $P$ that describes the graph $G = (\mathcal{V}, \mathcal{E})$. At lines 3–8, the algorithm iteratively refines $P$ into a more general one. At line 4, it enumerates possible generalizations of $P$ using the *Enumerate* function, defined as

$$Enumerate(P) = \{P' \mid P \rightsquigarrow P'\},$$

where the mutation rule ($\rightsquigarrow$) is defined in Figure 9. Intuitively, *Enumerate* mutates $P$ either by removing a node or edge variable (i.e., *RemoveNode* or *RemoveEdge*) or by widening the interval of a variable (i.e., *GeneralizeNode* or *GeneralizeEdge*). The operation *GeneralizeItv* widens an interval as follows:

$$GeneralizeItv(<[a_1, b_1], [a_2, b_2], \ldots [a_k, b_k]>) =$$
$$\{<[a'_1, b'_1], [a'_2, b'_2], \ldots [a'_k, b'_k]> \mid j \in [1, k], \forall i \neq j.\ a'_j = -\infty, b'_j = b_j, a'_i = a_i, b'_i = b_i\} \cup$$
$$\{<[a'_1, b'_1], [a'_2, b'_2], \ldots [a'_k, b'_k]> \mid j \in [1, k], \forall i \neq j.\ a'_j = a_j, b'_j = \infty, a'_i = a_i, b'_i = b_i\}.$$

At line 5, the algorithm selects a program $P'$ from the set of enumerated GDL programs $\mathcal{P}$ using the *Choose* function, defined as

$$Choose(\mathcal{P}, \mathcal{D}, y, Score) = \begin{cases} \bot & \text{if } \mathcal{P} = \emptyset \\ \arg\max_{P \in \mathcal{P}} Score(P, \mathcal{D}, y) & \text{otherwise} \end{cases}$$

Intuitively, the *Choose* function returns a better-scored program from the set of enumerated GDL programs $\mathcal{P}$. At line 6, it checks whether the refined program $P'$ has a higher score than the previous program $P$. If so, the algorithm continues refining. Otherwise (i.e., the refinement degrades the score), the algorithm returns the previous program $P$.

***Running Example.*** Suppose the four graph $G_1$, $G_2$, $G_3$, and $G_4$ in Figure 1a, 1b, 1c, and 1d are given as the training data (i.e., $\mathcal{D} = \{(G_1, 1), (G_2, 2), (G_3, 1), (G_4, 2)\}$) and the given graph $G$ is $G_3$ with label 1 (i.e., $y = 1$).

$$\frac{\delta_{\mathcal{V}} = (x, \_) \in \overline{\delta} \qquad D_x = \{\delta_{\mathcal{E}} \in \overline{\delta} \mid \delta_{\mathcal{E}} = (x, \_, \_) \vee \delta_{\mathcal{E}} = (\_, x, \_)\}}{\overline{\delta} \rightsquigarrow \overline{\delta} \setminus \{\delta_{\mathcal{V}}\} \setminus D_x} \; RemoveNode \qquad \frac{\delta_{\mathcal{E}} \in \overline{\delta}}{\overline{\delta} \rightsquigarrow \overline{\delta} \setminus \{\delta_{\mathcal{E}}\}} \; RemoveEdge$$

$$\frac{\delta_{\mathcal{V}} = (x, <\overline{\phi}>) \in \overline{\delta} \quad <\overline{\phi}'> \in GeneralizeItv(<\overline{\phi}>) \quad \delta'_{\mathcal{V}} = (x, <\overline{\phi}'>)}{\overline{\delta} \rightsquigarrow \overline{\delta} \setminus \{\delta_{\mathcal{V}}\} \cup \{\delta'_{\mathcal{V}}\}} \; GeneralizeNode \qquad \frac{\delta_{\mathcal{E}} = (x, y, <\overline{\phi}>) \in \overline{\delta} \quad <\overline{\phi}'> \in GeneralizeItv(<\overline{\phi}>) \quad \delta'_{\mathcal{E}} = (x, y, <\overline{\phi}'>)}{\overline{\delta} \rightsquigarrow \overline{\delta} \setminus \{\delta_{\mathcal{E}}\} \cup \{\delta'_{\mathcal{E}}\}} \; GeneralizeEdge$$

Figure 9: One-step mutation rules ($\rightsquigarrow$) for *Enumerate* function

Table 2: Statistics of the nine datasets

|  | MUTAG | Mutagenicity | BBBP | BACE | ENZYMES | PROTEINS | PTC | NCI1 | BA2Motif |
|---|---|---|---|---|---|---|---|---|---|
| # Graphs | 188 | 4337 | 2039 | 1513 | 600 | 1113 | 344 | 4110 | 1000 |
| # Avg nodes | 17.9 | 30.3 | 24 | 34 | 32.6 | 39.0 | 14.2 | 29.8 | 25.0 |
| # Avg edges | 19.7 | 30.7 | 25.9 | 36.8 | 62.1 | 72.8 | 14.6 | 32.3 | 25.4 |
| # Labels | 2 | 2 | 2 | 2 | 6 | 2 | 2 | 2 | 2 |
| # Node features | 1 | 1 | 9 | 9 | 19 | 2 | 1 | 1 | 1 |
| # Edge features | 1 | 1 | 3 | 3 | 0 | 0 | 1 | 0 | 0 |

1. The algorithm first transforms the input graph $G$ into a GDL program $P$ using the *Initialize* function. The following shows the given graph $G_3$ and a graphical representation of the transformed GDL program returned from *Initialize*($G_3$).

$$G_3 \quad = \quad \boxed{\langle 3.0 \rangle} \rightarrow \boxed{\langle 2.0 \rangle} \rightarrow \boxed{\langle 1.0 \rangle} \rightarrow \boxed{\langle 1.0 \rangle}$$

$$P \quad = \quad \boxed{\textbf{node } x \; \langle[3.0, 3.0]\rangle} \rightarrow \boxed{\textbf{node } y \; \langle[2.0, 2.0]\rangle} \rightarrow \boxed{\textbf{node } z \; \langle[1.0, 1.0]\rangle} \rightarrow \boxed{\textbf{node } h \; [1.0, 1.0]}$$
$$\textbf{edge }(x,y) \qquad \textbf{edge }(y,z) \qquad \textbf{edge }(z,h)$$

Suppose $\epsilon$ is 1. Then the score of the program $P$ is $Score(P, \mathcal{D}, 1) = \frac{|\{G_3\}|}{|\{G_3\}| + \epsilon} = \frac{1}{2}$.

2. In each iteration, the algorithm enumerates generalized GDL programs from the given GDL program $P$ using the *Enumerate* function, which removes node or edge variables or widens intervals of variables. The following shows a mutated GDL program $P'$ from the above GDL program $P$ (the node variable h and corresponding edge variable are removed).

$$P' \quad = \quad \boxed{\textbf{node } x \; \langle[3.0, 3.0]\rangle} \rightarrow \boxed{\textbf{node } y \; \langle[2.0, 2.0]\rangle} \rightarrow \boxed{\textbf{node } z \; \langle[1.0, 1.0]\rangle}$$
$$\textbf{edge }(x,y) \qquad \textbf{edge }(y,z)$$

The algorithm then selects the best-scored program from the set of enumerated GDL programs $\mathcal{P}$ using the *Choose* function. Suppose $P'$ is chosen. The score of the refined program $P'$ remains $Score(P', \mathcal{D}, 1) = \frac{1}{2}$.

3. The algorithm repeats this refinement process until all enumerated programs have a lower quality score than the previous program $P$. It then returns the best-scored program $P$.

## C   DATASET STATICTICS

Table 2 reports the statistics of the nine graph classification datasets. The row # Graphs indicates the number of graphs in each dataset. For example, BA2Motif contains 1000 graphs, which are split into train/validation/test sets with an 8:1:1 ratio (i.e., 800/100/100). The rows # Avg nodes and # Avg edges indicate the average number of nodes and edges per graph in each dataset, representing the average graph size. For instance, graphs in the PROTEINS dataset (# Avg nodes = 39.0 and # Avg edges = 72.8) are generally larger than those in the MUTAG dataset (# Avg nodes = 17.9 and # Avg edges = 19.7). The row # Labels shows the number of class labels in each dataset. For example, ENZYMES has six labels, whereas the other datasets have two. # Features and # Edge features show the number of node features and edge features in each dataset.

Table 3: Hyperparameter space for training the Baseline GNNs

| Hyperparameter | Space |
|---|---|
| Pooling | {MaxPool, MeanPool} |
| learning rate | {0.01, 0.005, 0.0005} |
| Hidden dimension | {20, 32, 64, 128} |
| Weight decay | {0, 1e-3, 5e-4, 5e-5} |
| Num layers | {3} |
| Dropout | {0.5} |
| Max epoch | {500} |
| Stopping patience | {100 epoch} |

Table 4: Hyperparameter space for training GDLNN

| Hyperparameter | Space |
|---|---|
| $\epsilon$ | {0.1, 0.01, 0.01 * $|\mathcal{D}|$} |
| $k$ | {0.01 * $|\mathcal{D}|$, 0.2 * $|\mathcal{D}|$, 0.4 * $|\mathcal{D}|$, 0.6 * $|\mathcal{D}|$, 0.8 * $|\mathcal{D}|$, $|\mathcal{D}|$} |
| learning rate | {0.01, 0.005, 0.0005} |

## D  HYPERPARAMETER SPACE

Table 3 shows the hyperparameter space used for training the baseline GNNs. We consider two pooling operations: MaxPool and MeanPool. The search space includes learning rates {0.01, 0.005, 0.0005}, hidden dimensions {20, 32, 64, 128}, weight decay values {0, 1e-3, 5e-4, 5e-5}, number of layers {3}, dropout rates {0.5}, maximum epochs {500}, and early stopping patience {100 epochs}. For MLP training, we select a subset of the hyperparameter space defined in Table 3.

Table 4 presents the hyperparameter space used for training GDLNN. For the GDL-layer, we choose $\epsilon$ and $k$ from the search space in Table 4. When training the MLP layers of GDLNN, we choose the learning rate in {0.01, 0.005, 0.0005}.

## E  EXPLAINABILITY AND ACCUMULATED COST COMPARISON

In this section, we present the explainability and cost comparison of GDLNN against the baseline methods that were omitted from the main paper due to space limitations.

***Qualitative comparison.*** Figure 10 presents the explanations of GDLNN, SubgraphX, and PL4XGL for classifying BA-2Motifs graphs into label 2. Again, all three methods correctly identify the motif associated with label 2. Figure 10a shows the subgraph explanation generated by GDLNN, which precisely captures the motif with five nodes. Similarly, SubgraphX (Figure 10b) provides a subgraph explanation that includes the five-node motif. PL4XGL (Figure 10c) provides a GDL program describing a property of the motif.

***Quantitative comparison.*** The two metrics *Sparsity* (higher is better) and *Fidelity* (lower is better) are defined as follows:

$$Sparsity : \frac{1}{N} \sum_{i=1}^{N} \left( 1 - \frac{|m_i|}{|M_i|} \right) \qquad Fidelity : \frac{1}{N} \sum_{i=1}^{N} \left( \mathbb{I}(\hat{y}_i = y_i) - \mathbb{I}(\hat{y}_i^{m_i} = y_i) \right)$$

In the above equations, $N$ indicates the number of test graphs, $m_i$ denotes the number of nodes in the subgraph explanation for the $i$'th graph, and $M_i$ is the number of nodes in the original $i$'th test graph. That is, a smaller subgraph explanation has a higher *Sparsity* score. The intuition behind the *Fidelity* score is that the predictions of the model should remain the same when the provided subgraph explanation is given. $\mathbb{I}(\hat{y}_i = y_i)$ is the indicator function that returns 1 if the $i$'th graph

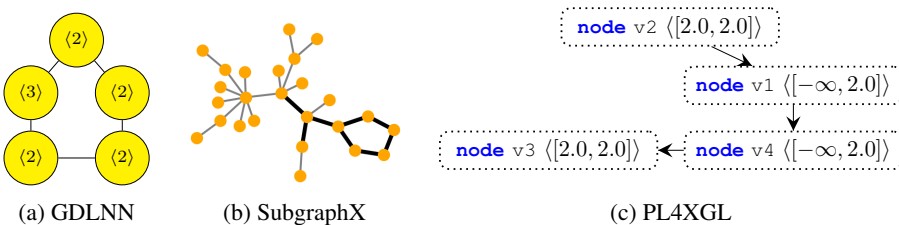

|       |       |       |
|-------|-------|-------|
| (a) GDLNN | (b) SubgraphX | (c) PL4XGL |

Figure 10: Provided explanations for graphs classified into label 2

is correctly classified by the model, and 0 otherwise. $\hat{y}_i^{m_i}$ is the prediction of the model when the subgraph explanation is given to the model. That is, remaining the same prediction makes the fidelity score lower.

Table 11 shows the balance between sparsity and fidelity of the explanations of GDLNN, SubgraphX, and PL4XGL. Again, the blue and orange lines present the sparsity and fidelity score of SubgraphX and GDLNN, respectively, while the gray dots present the sparsity and fidelity score of PL4XGL. As shown in Table 11, GDLNN consistently outperforms GIN+SubgraphX. Except for the BBBP dataset, GDLNN demonstrates a strictly better balance between sparsity and fidelity than GIN+SubgraphX. In comparison with PL4XGL, GDLNN performs worse than PL4XGL, since PL4XGL is an inherently explainable graph learning method (i.e., its fidelity score is guaranteed to be 0 Jeon et al. (2024)). By sacrificing some explainability, however, GDLNN achieves better accuracy than PL4XGL as shown in Table 1.

***Cost Comparison.*** Table 12 reports the accumulated cost (i.e., training + inference + explanation) of GDLNN, SubgraphX, and PL4XGL. The x-axis represents the number of classified and explained graphs, while the y-axis represents the accumulated cost. The results for the seven datasets are consistent with those of the two datasets presented in the main paper (Figure 8). Initially (i.e., training cost), the accumulated cost of GDLNN is lower than that of SubgraphX due to the expensive GDL program mining procedure. In terms of the overall accumulated cost, however, GDLNN is more efficient than SubgraphX. Except for the BA-2Motifs dataset, the total accumulated cost of GDLNN remains lower than that of SubgraphX. When compared to PL4XGL, the difference in accumulated cost is negligible, as both methods share the same training cost. For the Mutagenicity and NCI1 datasets, the total accumulated cost of GDLNN is slightly higher than that of PL4XGL because of its subgraph generation procedure (Algorithm 2).

## F  LLM USAGE IN THIS WORK

In this work, we primarily use LLMs to assist in writing the paper. We used LLMs to check for typos and grammatical errors. LLMs also revised some terms and sentences to enhance readability. Additionally, we used LLMs to more efficiently work with the existing codebase (e.g., using LIME) in our experiments.

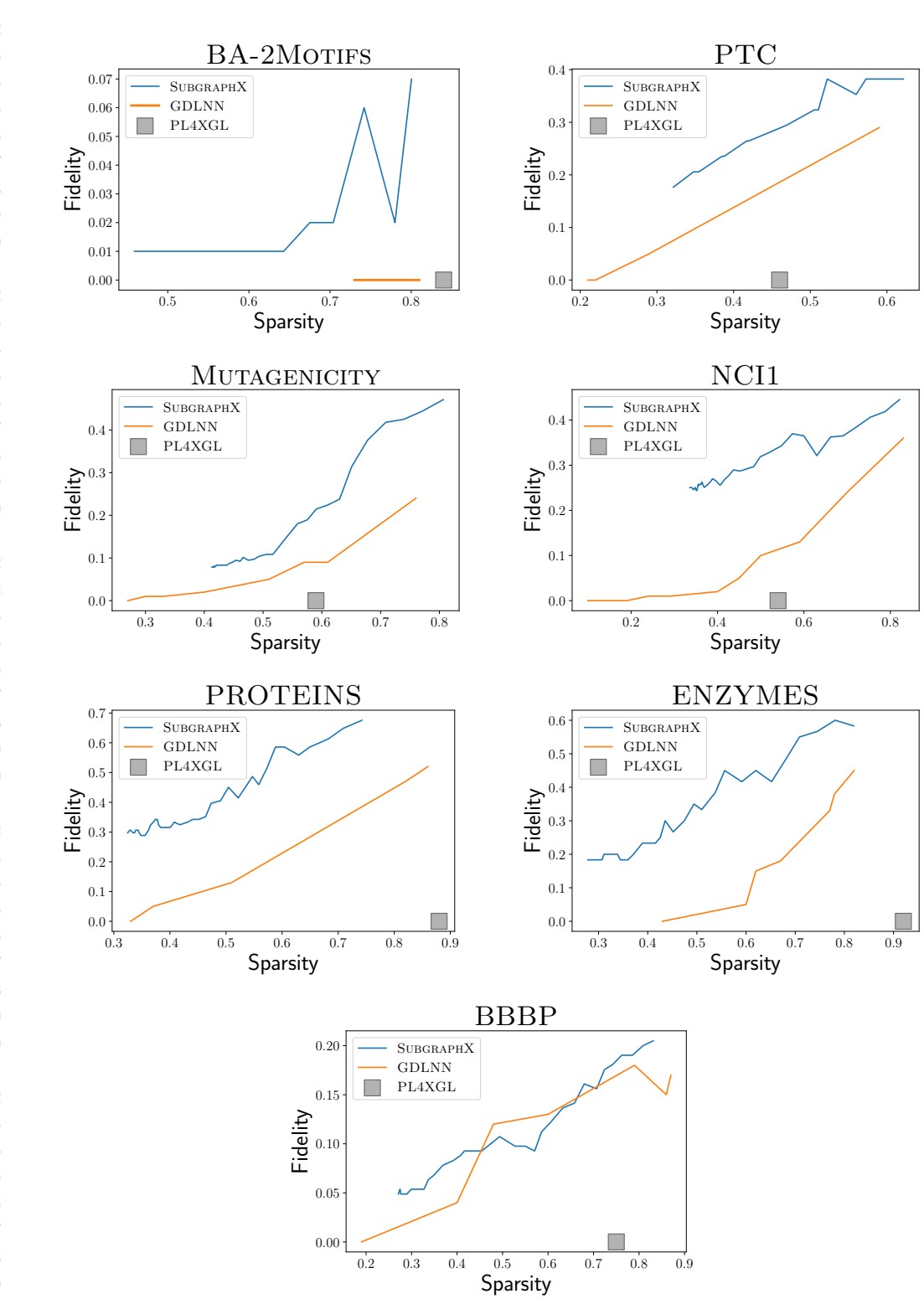

Figure 11: Balance between sparsity and fidelity comparison for seven datasets.

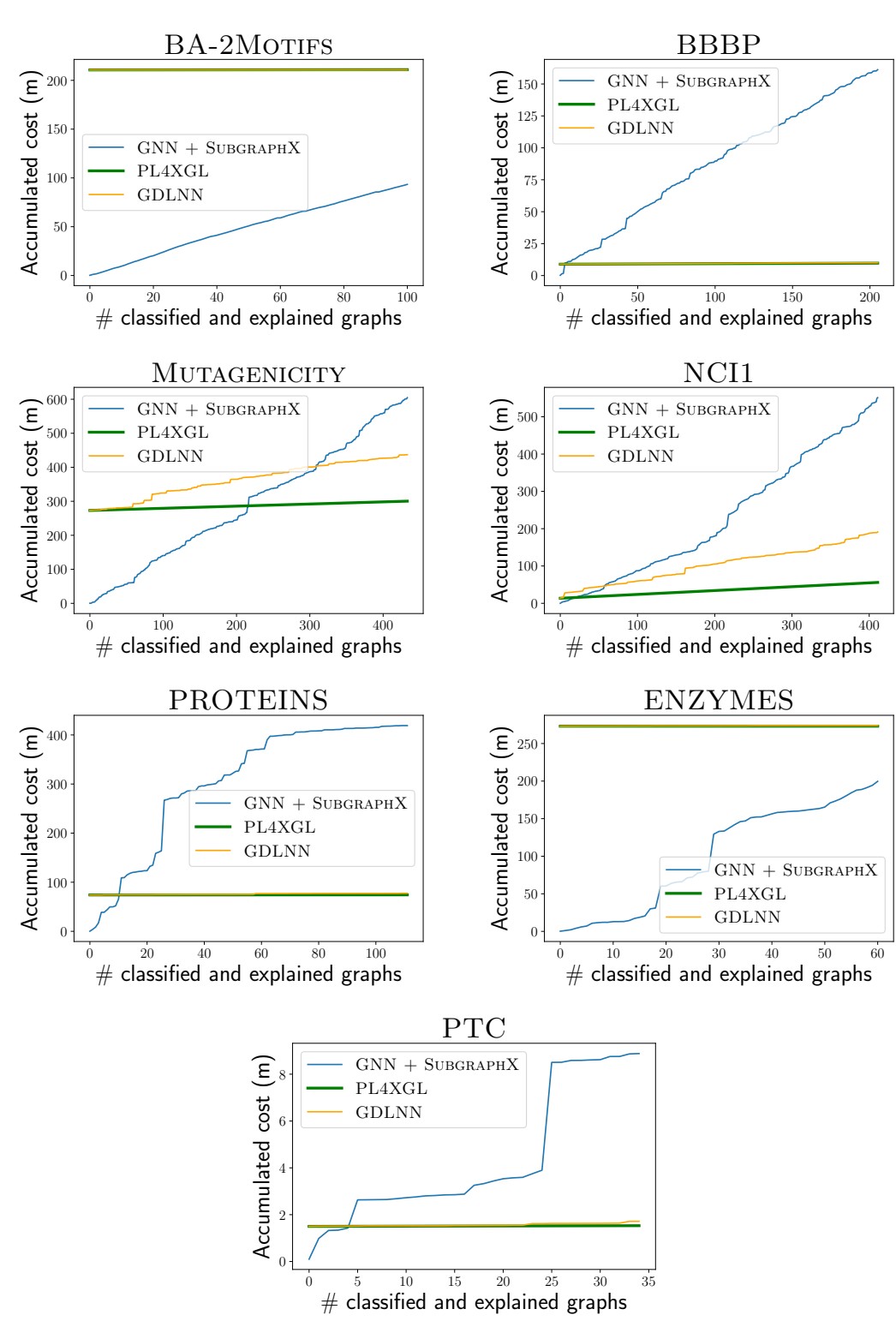

Figure 12: Accumulated cost comparison for seven datasets.

