# OpenReview forum: "GDLNN: Marriage of Programming Language and Neural Networks for Accurate and Easy-to-Explain Graph Classification"
_ICLR.cc/2026/Conference — ICLR 2026 Conference Withdrawn Submission_

### Official Review · Reviewer_kQjM · 2025-10-27

**Soundness:** 3
**Presentation:** 2
**Contribution:** 2
**Rating:** 6
**Confidence:** 4

**Summary:**

This paper's model, GDLNN, first mines a bunch of readable subgraph patterns (called GDL programs). It then checks which patterns a graph contains, turns that into a binary vector (like a checklist), and feeds that vector to an MLP for classification.
Since each feature is tied to a concrete pattern, the paper argues that they can explain predictions by ranking these programs (with LIME/SHAP) and showing the supporting subgraph. After reading and checking the paper and appendix, I give a **"weak accept" (6) rating and I think using auditable patterns as features is a reasonable idea for interpretability, but I still have some concerns about their evaluation and clarity.**

**Strengths:**

1. **Readable features.** Each dimension in the learned representation literally means “this specific typed subgraph exists in $G$.” This makes the model’s reasoning easy to audit and communicate.
2. **Practical explanation path.** The paper shows a nice workflow: rank important programs with LIME, then prune the graph to the minimal subgraph that still satisfies those programs. This gives sparse, faithful subgraphs at a lower cost. And the paper argues that once the GDL programs are mined, explanations for many new graphs are cheap.

**Weaknesses:**

1. **Novelty positioning.** Conceptually, GDLNN mines discriminative GDL programs (very similar in spirit to PL4XGL) and then feeds their binary activations to an MLP. That is a reasonable and useful engineering step, but the paper currently frames this as a new paradigm. I think the author could use a more nuanced and honest positioning that acknowledges prior work more clearly.

2. **No end-to-end training.** The GDL programs are mined once, then frozen. The activation $\sigma(P,G)$ is a hard 0/1 test of pattern presence. The MLP never updates the programs. This is fine, but it should be stated clearly as a two-stage pipeline instead of implying a single differentiable model. The appendix hints at a richer count-based $\sigma'(P,G)$ but it seems not used in the main experiments.

3. **Reproducibility details and inconsistency.** A few hyperparameter details are inconsistent between main text and appendix (e.g., candidate values for $\epsilon$, how $k$ is chosen). In Sec 5, the paper defines$k$ as the number of kept programs and says “we select $k$ from ${0.01\cdot|D|, 0.2\cdot|D|, 0.4\cdot|D|, 0.6\cdot|D|, 0.8\cdot|D|, |D|}$ based on the validation set,” together with the TopK definition $\text{TopK}(P)={P \in \mathcal{P} \mid |{P' \in \mathcal{P} \mid \text{Score}(P',D,y)\ge \text{Score}(P,D,y)}|\le k}$. However, Appendix D says that both $k$ and $\epsilon$ are tuned on validation, and it lists the search grid for $\epsilon$ as ${0.1, 0.01, 0.01\cdot|D|}$, which is not stated in the main text. The appendix also implies that $\epsilon$ directly affects the mining score $\text{Score}(P,D,y)=\frac{|{G\in D_y\mid G\models P}|}{|{G\in D\mid G\models P}|+\epsilon}$,but the main text not clearly reports which $\epsilon$ was finally used per dataset.

4. **Explainability robustness.** Explanation quality is evaluated mainly with LIME. LIME can be unstable on sparse binary features. However this paper didn't provide either a short sensitivity study or a comparison with SHAP to check that the fidelity and sparsity curves are not an artifact of one attribution method.

**Questions:**

1. Can you provide ROC-AUC and a scaffold/OOD split for the molecule datasets, or at least discuss how close train and test graphs can be? Random splits may leak near-duplicate structures across train and test, which can inflate accuracy.

2. On MUTAG you report $100%$ accuracy with zero variance. Can you provide evidence that this is not just memorizing a few motifs tied to specific labels? For example, how does performance look if you hold out whole motifs or use fewer programs?

3. When you sweep $k$ (number of kept programs) and $\epsilon$, do you rerun program mining, or do you mine once and then just select top-$k$? I found that the appendix and main text list slightly different and inconsistent $\epsilon$ grids as I mentioned above. Which one matches the reported numbers?

4. The cost comparison argues that explanations are cheap once the program bank is mined. Suppose I deploy to a new dataset or a new label space. Do I have to re-mine programs, or can I reuse the old bank?

---

### Official Review · Reviewer_LQ7h · 2025-10-31

**Soundness:** 3
**Presentation:** 2
**Contribution:** 1
**Rating:** 2
**Confidence:** 3

**Summary:**

This paper proposes GDLNN, a two-stage graph classification architecture that
first mines a set of graph patterns expressed in a domain-specific Graph De-
scription Language (GDL), then encode each graph as a binary feature vector
indicating the presence of each mined program and feed this vector into an MLP
classifier. A graph ”activates” a program if it contains a subgraph matching that
pattern. Explanations are obtained post-hoc: the authors instantiate LIME to
rank programs and then apply a refinement routine to extract a subgraph that
preserves the selected programs.

**Strengths:**

* The overall architecture is clear and intuitive: mined programs become
binary features, and a standard MLP operates on them for the final prediction.
* The method attains strong accuracy while still enabling post-hoc explana-
tions.

**Weaknesses:**

* (Most important) Novelty is very limited and incremental. The method
proposed adopts the mining process from PL4XGL, and the explainability is
obtained through already established post-hoc techniques. The only novelties
are the use of top-k programs as binary features instead of only taking the best
program (PL4XGL), the addition of an MLP for prediction and the refinement
step to get the post-hoc explanation.

* Cost analysis and discussion are scarce, and potentially misleading. In
the appendix, costs are reported for 7 of the 9 datasets tested (BACE dataset is
shown in the main paper; it’s unclear why MUTAG is omitted here). Although
GDLNN has lower accumulated cost on 6/8 datasets (MUTAG missing), it is
markedly higher than GNN+SubgraphX on BA-2Motifs and ENZYMES, with
no further analysis on why this behaviour is obtained. Thus, despite the claim
of GDLNN being faster in the main paper, its cost appears to be highly dataset-
dependent.

* What value was assigned to k to get the accuracies in Table 1 is not
shown. The absence of this value k makes the assessment of the performance
difficult

* SubgraphX settings unclear. The user-defined upper-bound controlling
SubgraphX explanations size is not reported, making the interpretation of the
fidelity-sparsity plots difficult.

**Questions:**

* The method loses interpretability while it could retain some level of self-
explainability. Although the mined program features are inherently interpretable,
the unconstrained MLP obscures interpretability, forcing the adoption of post-
hoc explainability techniques. Why not avoid the MLP by adopting a non-
negative linear classifier that could help achieve some level of self-explainability?

* The authors claim that existing model explanation techniques can be used,
yet only LIME is evaluated. Why LIME over alternatives like SHAP or gradient-
based feature attribution? Please justify and possibly compare.

* The authors define the accumulated cost as the sum of the training, infer-
ence, and explanation times, but separating these components would be more
informative to understand the costs of each phase (the authors themselves note
substantial differences regarding training time). Please justify this decision.

---

### Official Review · Reviewer_AQG2 · 2025-11-01

**Soundness:** 3
**Presentation:** 3
**Contribution:** 3
**Rating:** 6
**Confidence:** 3

**Summary:**

This paper introduces GDLNN, a hybrid architecture that integrates a domain-specific programming language (Graph Description Language, GDL) with neural networks for graph classification. The core idea is to generate interpretable and expressive graph representations using GDL programs, which are then classified using an MLP. The authors claim that this approach enhances both accuracy and interpretability compared to conventional GNN-based methods. Experiments on nine benchmark datasets demonstrate strong performance in accuracy, explainability (measured via LIME), and computational efficiency when explanation cost is considered.

**Strengths:**

1. Novel integration of symbolic and neural paradigms: The proposed combination of a programming language (GDL) and neural network is innovative and provides a promising direction for interpretable graph learning.

2. Explainability focus: The authors convincingly demonstrate that GDLNN enables model-agnostic explanation methods (e.g., LIME, SHAP) to directly interpret its predictions.

3. Efficiency consideration: Including explanation cost in total model evaluation is insightful and practically relevant.

4. Well-structured evaluation: The paper provides both qualitative and quantitative analyses, with visual examples and detailed metrics (fidelity, sparsity).

**Weaknesses:**

While the paper claims that GDLNN can be combined with any post-hoc explanation method to generate interpretable results, this statement seems only partially true. In practice, the GDL layer produces discrete binary representations (each feature indicates whether a GDL program matches a given graph), which are not differentiable. Therefore, gradient-based explainers such as Integrated Gradients, Saliency Maps, or Grad-CAM cannot be directly applied. The approach is thus mainly compatible with perturbation-based, model-agnostic explainers (e.g., LIME, SHAP, Anchors). The paper should clarify whether GDLNN can meaningfully cooperate with gradient-based explanation methods, and if not, discuss this limitation explicitly.


In addition, the mechanism by which GDLNN works with LIME is not clearly explained. Algorithm 2 abstractly defines the step ImportantFeatures(G, M, T) but does not describe how perturbations are generated in the discrete binary feature space or how the identified important features are mapped back to subgraph structures. More details are needed on how LIME interacts with the GDL layer, including the perturbation sampling strategy and the reconstruction of subgraph explanations.


Finally, the comparison of explainability methods is incomplete. The experiments only evaluate SubgraphX and PL4XGL, omitting well-established baselines such as GNNExplainer, PGExplainer, and GraphLIME. Including these would provide a more comprehensive and fair comparison, and help position GDLNN within the broader landscape of explainable graph learning.

**Questions:**

See Weaknesses.

---

### Official Review · Reviewer_N6o9 · 2025-11-03

**Soundness:** 3
**Presentation:** 3
**Contribution:** 2
**Rating:** 4
**Confidence:** 3

**Summary:**

This paper presents GDLNN, a hybrid architecture that integrates a domain-specific GDL with neural networks for graph classification. The GDL component encodes explicit graph patterns as symbolic programs, while a MLP operates on the binary activations of these patterns to produce predictions. The core motivation is to create graph representations that are both interpretable and expressive, addressing the limitations of GNNs, whose learned embeddings are often opaque and entangled. The authors claim that the mined GDL programs yield interpretable representations that allow standard explainability methods (e.g., LIME, SHAP) to be applied directly. Experiments across nine benchmark datasets show that GDLNN achieves competitive or higher accuracy than GNN baselines (e.g., GIN, GCN, GAT) and that the combination of GDLNN with LIME provides better sparsity–fidelity trade-offs than SubgraphX explanations for GNNs.

**Strengths:**

The paper tackles an important and underexplored direction, symbolic-neural hybrid modeling for explainable graph learning, and presents an architecture that is both conceptually interesting and relatively simple to implement. Integrating a declarative language layer (GDL) for interpretable pattern extraction with a neural classifier is an appealing idea for bridging symbolic reasoning and representation learning. The presentation is thorough, with clear syntax and semantics definitions for GDL (Figures 2-3) and explicit mining algorithms (Algorithms 1 and 3). The use of interpretable intermediate representations allows off-the-shelf explainers like LIME or SHAP to operate without architectural modifications, which is a pragmatic design choice. Empirical evaluation is broad, covering diverse datasets and comparing to both symbolic (PL4XGL) and neural (GNN + SubgraphX) baselines.

**Weaknesses:**

- The paper’s technical novelty and contribution depth are modest. Most of the work builds directly on the authors’ previous PL4XGL (Jeon et al., 2024) system, which already introduced GDL as a symbolic approach to explainable graph learning. GDLNN primarily adds a shallow MLP layer and reuses the existing GDL mining and scoring pipeline with minimal conceptual changes. The claim that “GDLNN is the marriage of programming language and neural networks” oversells what is effectively a two-stage system—symbolic feature mining followed by standard supervised learning—without genuine co-training or mutual optimization between the symbolic and neural components.

- From a ML standpoint, the proposed system behaves like a feature-engineering pipeline rather than a unified differentiable model. The GDL program mining step is discrete, non-gradient-based, and computationally heavy, while the downstream MLP training is conventional. The lack of end-to-end optimization means the neural component cannot influence the symbolic search space, limiting expressiveness and adaptability. This disconnect also undermines the claimed synergy between programming languages and neural networks.

- On the theoretical side, the semantics and learning objectives of GDLNN are largely definitional rather than novel. The pattern-matching and scoring functions replicate existing approaches in symbolic subgraph mining or inductive logic programming, and the formal grammar of GDL provides syntactic clarity but no new learning capabilities. The “interpretability” argument is also overstated: while GDL programs are human-readable, their mined intervals and node-edge mappings can become complex, especially for large graphs, making practical interpretation nontrivial. Moreover, the paper treats applying LIME or SHAP on top of GDL features as evidence of explainability, but this does not demonstrate causal fidelity or human-aligned interpretability.

- Empirically, while GDLNN performs competitively on small to mid-scale datasets (Table 1, p.7), its advantage is not significant over strong GNN baselines. Gains on MUTAG and BACE are minor, while on larger datasets such as PROTEINS, GDLNN slightly underperforms. The improvement over PL4XGL seems primarily due to the neural classifier, not the hybrid design. The evaluation lacks variance analysis across random seeds, and scalability beyond thousands of nodes per graph is untested. The explanation metrics (Figure 7, p.8) rely on sparsity and fidelity alone, which are simplistic and insufficient to assess interpretability quality. The cost analysis (Figure 8, p.9) demonstrates marginal efficiency improvements once mining time is included.

- Finally, the paper’s writing and framing could be more critical and focused. Much of the space (Sections 4-5, Appendices B-D) is devoted to reproducing existing definitions and pseudocode from earlier GDL-based work, giving the impression of incremental advancement rather than a breakthrough. The discussion of limitations and future directions is minimal.

**Questions:**

- How does the proposed GDLNN differ fundamentally from PL4XGL beyond adding an MLP classifier? Can the MLP’s role be seen as a learned weighting over symbolic patterns rather than genuine neural integration?

- Could GDL program mining be jointly optimized with the neural loss via reinforcement learning or differentiable relaxation to realize an actual hybrid model?

- How scalable is the GDL mining process to larger graph datasets (e.g., OGB benchmarks)? Is there any parallelization or pruning mechanism?

- How interpretable are the mined programs in practice? Have human evaluators been consulted to assess their comprehensibility or domain relevance?

- Since LIME operates on tabular feature vectors, what guarantees exist that its explanations correspond to meaningful graph substructures rather than artifacts of binary encodings?

**Details Of Ethics Concerns:**

None.

---

### Author Response · Authors · 2025-11-18

We thank the reviewers for their feedback. We will take the comments under consideration as we improve the work for future submission.

---

### Note · Authors · 2025-11-18

I have read and agree with the venue's withdrawal policy on behalf of myself and my co-authors.